# Is Tempranillo Blanco Grapevine Different from Tempranillo Tinto Only in the Color of the Grapes? An Updated Review

**DOI:** 10.3390/plants11131662

**Published:** 2022-06-23

**Authors:** Tefide Kizildeniz, Inmaculada Pascual, Ghislaine Hilbert, Juan José Irigoyen, Fermín Morales

**Affiliations:** 1Universidad de Navarra, Plant Stress Physiology Group (Environmental Biology Department), Associated Unit to CSIC, EEAD, Zaragoza, Faculties of Sciences and Pharmacy, Irunlarrea 1, 31008 Pamplona, Navarra, Spain; tkizildeniz@alumni.unav.es (T.K.); ipascual@unav.es (I.P.); jirigo@unav.es (J.J.I.); 2EGFV, Université de Bordeaux, Bordeaux Sciences Agro, INRAE, ISVV, 33882 Villenave d’Ornon, France; ghislaine.hilbert@inrae.fr; 3Instituto de Agrobiotecnologia (IDAB), CSIC-Gobierno de Navarra, Avda. de Pamplona 123, 31192 Mutilva, Navarra, Spain

**Keywords:** Tempranillo Blanco and Tinto grapevine, response to climate change, fruit-bearing cuttings, temperature gradient greenhouses, differences in physiology, growth, grape production and quality

## Abstract

Tempranillo Blanco is a somatic variant of Tempranillo Tinto that appeared as a natural, spontaneous mutation in 1988 in a single shoot of a single plant in an old vineyard. It was vegetatively propagated, and currently wines from Tempranillo Blanco are commercially available. The mutation that originated Tempranillo Blanco comprised single-nucleotide variations, chromosomal deletions, and reorganizations, losing hundreds of genes and putatively affecting the functioning and regulation of many others. The most evident, visual change in Tempranillo Blanco is the anthocyanin lost, producing this grapevine variety bunches of colorless grapes. This review aims to summarize from the available literature differences found between Tempranillo Blanco and Tinto in addition to the color of the grapes, in a climate change context and using fruit-bearing cuttings grown in temperature-gradient greenhouses as research-oriented greenhouses. The differences found include changes in growth, water use, bunch mass, grape quality (both technological and phenolic maturity), and some aspects of their photosynthetic response when grown in an atmosphere of elevated CO_2_ concentration and temperature, and low water availability. Under field conditions, Tempranillo Blanco yields less than Tempranillo Tinto, the lower weight of their bunches being related to a lower pollen viability and berry and seed setting.

## 1. An Introduction to Tempranillo Blanco, a New Variety That Originated from Tempranillo Tinto

Tempranillo Blanco is a new green-yellow variety of Tempranillo Tinto that was discovered in a centenary vineyard in 1988 [1], and only recently, a limited number of ha (607 ha) under exploitation makes wines from this variety commercially available. Tempranillo Blanco resulted from genome changes that include chromosomal deletions, reorganizations between chromosomes and gene single-nucleotide variations, whose consequences in plant functioning and aspects of vegetative growth and yield require investigation. The most evident, visual change in Tempranillo Blanco is the anthocyanin lost, losing color in the grapes.

From a genetic point of view, Tempranillo Blanco is the result of a spontaneous, catastrophic genome alteration that resulted in 313 hemizygous genes because of right arm deletions of linkage groups 2 and 5 [2]. Additionally, intra-chromosomal and inter-chromosomal translocations between three linkage groups (linkage groups 2, 5, and 9) flank the deleted fragments [2]. As a consequence, the functional copy for the MYB transcription factors was lost, affecting berry color, and gamete viability was impaired, which might lower production through berry set [2,3]. This suggests that other loci different from those related to grape color have been affected, losing or affecting hundreds of genes.

From an ampelographic point of view, differences between Tempranillo Blanco and Tinto are scarce, except the color of the berries, green-yellow, and blue-black, respectively. Tempranillo Blanco leaves are smaller than those of Tempranillo Tinto, similar in form, with greater swelling of the adaxial side and greater density of trichomes on abaxial veins. In Tempranillo Blanco, the cluster is medium-sized and loose (large in size in Tempranillo Tinto), and the berry has a slightly flattened shape (spherical in Tempranillo Tinto). There are also differences in the number of seeds per berry and in seed length and width, with Tempranillo Blanco having a lower number of seeds but being larger in size (higher length and width) [1,4].

Lower bunch size in Tempranillo Blanco than in Tempranillo Tinto was related to an impaired pollen viability and berry and seed setting [3,5]. Additionally, pre-flowering climatic factors impact the reproductive traits of Tempranillo Blanco. Tello et al. [3] reported an increased number of seedless berries after cold days at pre-flowering and rainfalls at flowering, whereas pollen viability and number of seeds per berry were rather unaffected by these environmental conditions in Tempranillo Blanco.

Grapes from Tempranillo Blanco, as above mentioned, are green-yellow. Grapes mature early, and so wines can reach a high alcohol content (≥13.3% in volume). The total acidity content of both grapes and wines is also high. In terms of acid concentration, tartaric acid has medium-high values, however, their contribution to wine acidity is low and limited by the potassium levels present, which are usually high. Acidity is therefore determined by the high concentration of malic acid, which strengthens wine freshness, a characteristic considered positive at sensory level. Sensory analysis of Tempranillo Blanco wines shows high-quality organoleptic characteristics. In Tempranillo Blanco wines, the concentration of fruity aromas of a fermentative character is high, highlighting some alcohol acetates, such as isoamyl acetate (banana aroma), with a high incidence in the aroma, and hexil acetate (pear aroma). Likewise, ethyl esters of fatty acids are also abundant in wines, mainly ethyl-3-hydroxybutyrate and ethyl butyrate (aromas of pineapple, kiwi, and strawberry) [6].

## 2. The Fruit-Bearing Cutting Technique as a Tool to Investigate Differences between Tempranillo Blanco and Tinto

Different methods have been used to evaluate differences between Tempranillo Tinto and Tempranillo Blanco. This review includes genetic approaches, agronomic characterization and evaluation of the resulting wines under field conditions, response to climatic variables both in the field and under fully controlled conditions in greenhouses, and response to simulations of future, foreseen conditions related to climate change. Fruit-bearing cutting is an experimental approach that allows growing a grapevine plant in a pot, which facilitates its management and where abiotic stress factors can be applied either individually or in combination in a fully controlled way.

In this review, data from Tempranillo Blanco and Tinto fruit-bearing cuttings were used. *Vitis vinifera* L. cv. Tempranillo Tinto (RJ-43 clone) and Tempranillo Blanco (CI-101 clone) dormant cuttings were obtained from an experimental vineyard of the ICVV (Institute of Sciences of Vine and Wine, Logroño, La Rioja, Spain). We used clone RJ-43 as one of the most widely grown Tempranillo Tinto clones. In fact, genetic comparisons between Tempranillo Tinto and Blanco were made using clone RJ-51 as a reference, representative, and widely cultivated clone in La Rioja [2,3]. In the case of Tempranillo Blanco, we used the only clone available as did Carbonell-Bejerano et al. [2] and Tello et al. [3].

Fruit-bearing cuttings were obtained according to Mullins [7], as described in detail in Morales et al. [8]. In brief, indole butyric acid (300 mg L^−1^) was used to root the cuttings. Cuttings were placed in a rock-wool heat-bed (27 °C) inside a cool room (4 °C). The rooted-cuttings, one month later, were planted in 0.8 L plastic pots (in a mixture of sand, perlite, and vermiculite (1:1:1, in volume)) and placed in the pre-culture greenhouse. On each plant, we allowed only a single flowering stem to develop, which resulted in one cluster per plant. Vegetative growth was controlled until fruit set by manual pruning, maintaining four leaves per plant. When plants reached fruit-set stage, they were transplanted to 13 L plastic pots (in a mixture of peat and perlite (2:1, *v*/*v*)). Pre-culture greenhouse conditions were as follows: 26/15 °C and 60/80% relative humidity (RH) (day/night) and 15 h photoperiod. Natural sunlight was supplemented with 500 µmol m^−2^ s^−1^ PPFD (photosynthetic photon flux density) at inflorescence level with high-pressure metal halide lamps (OSRAM^®^, Augsburg, Germany). Lamps were triggered when sunlight dropped below 900 µmol m^−2^ s^−1^. Nutrient solution was that described by Ollat et al. [9]. After fruit set, treatments were applied after transferring plants to the below-described temperature gradient greenhouses. Growth/success rate of Tempranillo Tinto and Tempranillo Blanco fruit-bearing cuttings was similar.

## 3. The Temperature Gradient Greenhouses Simulate Future Climate Conditions for Growing Tempranillo Blanco and Tinto

Under field conditions, sometimes is difficult to identify which factor is causing a specific response. In addition, in a context of climate change, simulating future climatic conditions in the field is not easily affordable and economically expensive. This trouble can be solved using the approach of growing plants under fully controlled conditions. Today, climate change can be simulated using greenhouses [10]. In the present work, we used temperature gradient greenhouses (TGG) for growing Tempranillo Blanco and Tinto and to investigate their physiological response to climate change. TGGs allow to manage climatic variables (temperature, water availability, atmospheric CO_2_ concentration, etc.) comparing current and foreseen conditions and to assign a physiological response to one specific factor [10]. TGG are based on temperature gradient tunnels [11]. Briefly, TGG are permanently fixed to the soil, have aluminum structure with walls of polycarbonate and polyethylene roof. Readers are referred to Morales et al. [10] for details on how to create the temperature gradient within the modules of the TGG, injection of CO_2_ from CO_2_ cylinders and water stress control.

Treatments started one week after fruit set and ended at maturity (berries with 22°Brix) and were as follows: ambient temperature versus ambient +4 °C; 400 μmol CO_2_ mol^−1^ (ppm) versus 700 ppm. Additionally, substrate water content was monitored by placing sensors into the pots (EC-5 Soil Moisture Sensors, Decagon Devices Inc., Pullman, WA, USA). Ten cuttings (10 cuttings × 2 cultivars × 2 CO_2_ concentrations × 2 temperatures × 2 water availabilities) were used per combination. We used 4 TGG, two at elevated CO_2_ and two at current one. On each TGG, one module of elevated temperature and one at ambient temperature were used. On each module, 20 plants were grown (10 of Tempranillo Blanco and 10 of Tempranillo Tinto), of which half of them were fully irrigated and the other half were water stressed. Therefore, 160 cuttings in total were used per experiment.

Under full irrigation, plants were maintained at 300–400 g H_2_O L^−1^ substrate) [12]. Pot water content fluctuated, despite the fact that the design was made to minimize water evaporation from the substrate [10], possibly related to evapotranspiration (see Kizildeniz et al. [13] for further discussion). Water stress (withholding irrigation) treatments lowered pot water content to 0–100 g H_2_O L^−1^ substrate [12]. At this point when tendrils and leaves had lost turgor, plants were re-irrigated ensuring plant viability. This management resulted in a cyclic drought of 7 irrigations in 3 months. Nutrient solution or water were used, adjusting nutrients to the amount that had received full-irrigated plants in that cycle. Cycles lasted two weeks (first cycle) and then one week (rest of the cycles) [13,14]. Representative records of pot water contents from fruit set to maturity are shown in Kizildeniz et al. [13,14]. Water management in these experiments was based on experience acquired working with Tempranillo Tinto fruit-bearing cuttings and TGG [15].

Sampling was made at five phenological stages: (I) one week before veraison (around 60 days after flowering), (II) mid-veraison, (III) one week after mid-veraison, (IV) two weeks after mid-veraison, and (V) maturity (around 22°Brix). Roots, shoots, petioles, leaves, and whole clusters (berries and rachises) were sampled. Plants were maintained without pruning from fruit set to maturity allowing vegetation to grow freely.

## 4. Growth, Water Use, and Production of Tempranillo Blanco and Tinto Fruit-Bearing Cuttings under Simulated Climate Change Conditions

Pot substrate water capacity is the water retained by the substrate pores by capillarity once drainage had occurred after irrigation. This water is lost over time by direct evaporation and plant water absorption at root level. Data reported in a previous work indicate that Tempranillo Tinto consumed more water than Tempranillo Blanco [12]. At plant level, larger leaf area and higher density of roots contribute to such higher consumption in Tempranillo Tinto. At leaf level, a higher density of stomata may contribute. Leaf dry weight (DW) and plant root DW were higher in the Tempranillo Tinto cultivar when compared to Tempranillo Blanco, and leaf area per plant was lower in Tempranillo Blanco (especially under full irrigation and ambient temperature) [12]. Additionally, the number of stomata per unit leaf area is overall lower in Tempranillo Blanco than in Tempranillo Tinto, differences that were maximized when grown at elevated CO_2_ (Figure 1). Illustrative pictures from which these data were obtained are shown in Appendix A. Differences in stomatal dimensions, if any, were minor (Appendix A). It cannot be ruled out that in the genome of Tempranillo Blanco had loci altered by the mutation that refer to stomatal density. However, there were not loci/genes annotated as affecting stomatal traits when genomes of Tempranillo Blanco and Tinto were compared [2]. This hypothesis requires further investigation.

Whether water and nutrients are not limiting factors, grapevine growth [17] and yield are enhanced by elevated atmospheric CO_2_ concentration, but to a different extent among cultivars [14]. In line with previous reports [14,18,19], the impact of water stress was less intense when plants were grown in presence of elevated CO_2_. Possibly because of the increased content of water in the leaves of Tempranillo Blanco and Tinto (see below), vegetative growth (total mass) was more stimulated than reproductive growth (fresh weight (FW) of berries and rachises) by elevated CO_2_ [12]. For instance, elevated CO_2_ increased leaf DW by 22% and when combined with elevated temperature it reached 29%. Grapevine berry diameter is a trait that shows high variability [20]. In agreement with a previous report [21], berry diameter from plants grown under elevated CO_2_ was 8% larger than in those grown at ambient CO_2_. A new varietal difference could also be observed in terms of vegetative biomass. Whereas in Tempranillo Blanco elevated CO_2_ produced larger increases in leaf mass (30%), CO_2_ specially stimulated root biomass in Tempranillo Tinto (48%) [12]. Moreover, elevated CO_2_-mediated leaf DW increases were larger in droughted (45%) than in full irrigated (24%) plants. Final leaf DW values of droughted plants grown in presence of elevated CO_2_, however, did not reach those of control, full-irrigated plants. The enhanced growth observed in grapevine grown at elevated CO_2_ is likely a consequence of its high photosynthetic rates [21,22,23,24,25,26]. Stomatal conductance (see below) and transpiration rates [24,25,27,28] as well as stomatal density ([29,30]; Figure 1) are, however, reduced by elevated CO_2_ exposure, increasing water use efficiency [31].

According to our results, differences in reproductive performance between Tempranillo Blanco and Tinto are not so strong. We made a 3-year experiment growing Tempranillo Blanco and Tinto fruit-bearing cuttings in TGGs simulating future climate change conditions. As a whole and taking into account all the years and treatments, bunch FW of Tempranillo Blanco was 8% higher than that of Tempranillo Tinto [32]. No differences were found between cultivars in other reproductive performance-related traits such as berry diameter, relative skin mass, and number of seeds per berry [12]. Under field conditions, however, there are strong differences between the reproductive performance of Tempranillo Blanco and Tempranillo Tinto, starting from gamete viability and ending in strongly different yields [3]. Minor differences observed in the reproductive performance between these two cultivars in other works [32] should be related in some way to the conditions assayed using fruit-bearing cuttings and greenhouses for growing the plants.

## 5. Grape Quality of Tempranillo Blanco and Tinto Fruit-Bearing Cuttings under Simulated Climate Change Conditions

Grape quality is affected by many features (see, for a recent review, Poni et al. [33]). Experiments performed with TGGs showed that elevated temperature and CO_2_ modify berry characteristics [32,34,35,36]. Varietal differences detected between Tempranillo Blanco and Tinto with regards to grape quality are an important aspect of this report. Our previous experiments during three consecutive growing seasons indicate changes in sugars, organic acids, and polyphenols [32]. Compared with Tempranillo Tinto, Tempranillo Blanco showed higher grape total soluble solids (mainly sugars), lower acidity, and higher pH related to a decreased concentration of the organic acids malic and tartaric [32]. In addition, total polyphenol index was lower in Tempranillo Blanco than in Tempranillo Tinto, which was due to the absence of anthocyanins (resulting in grapes of green-yellow color) (Appendix A). Anthocyanins in Appendix A were determined following the procedure described in Acevedo De la Cruz et al. [37]. The concentration of flavonols was also lower in Tempranillo Blanco than in Tempranillo Tinto (Table 1). Moreover, when compared to Tempranillo Tinto, cluster FW and content of water in berries were higher in Tempranillo Blanco [32]. Results of reproductive performance, however, do not confirm those of Tello et al. [3]. Readers are referred to Morales et al. [8] for comparison of results obtained with grapevines grown in vineyards in the field with those of fruit-bearing cuttings grown in greenhouses, including the usefulness of the use of fruit-bearing cuttings to evaluate grapevine reproductive traits. These grape quality data may be used as the basis for further investigations on the differences between Tempranillo Blanco and Tinto from a genetic point of view.

## 6. Physiology of Tempranillo Blanco and Tinto: Gas Exchange Properties and Photosynthetic Acclimation

In this review, we have analyzed elevated temperature, elevated CO_2_ concentration, and a reduction in water for irrigation as three of the major factors related to climate change. Upon exposure to elevated CO_2_, within the first days, stomata close, transpiration decreases, but photosynthesis increases [39,40,41,42]. In oak and soybean, for instance, the enhanced C fixation mediated by elevated CO_2_ may last [43,44]. However, under longer exposure (weeks, months, or beyond), photosynthesis down-regulates (i.e., this response may be abolished, at least partly, by photosynthetic acclimation to elevated CO_2_, a phenomenon that decreases the plant photosynthetic capacity) [14,41,45,46,47,48,49,50]. Photosynthetic acclimation is evidenced measuring at a fixed CO_2_ concentration (either current or elevated) and comparing photosynthesis in plants grown at ambient and elevated CO_2_ [50]. In a recent work, Kizildeniz et al. [13] measured photosynthesis at 700 ppm CO_2_ in Tempranillo Blanco and Tinto plants that were grown at current and elevated CO_2_ concentration [13]. Decreases in photosynthesis were observed in both Tempranillo Blanco and Tinto, concluding that plants acclimated in response to the elevated CO_2_. Tempranillo Blanco had slightly higher photosynthetic rates than Tempranillo Tinto, but only at the phenological stages of one and two weeks after mid-veraison [13].

Another line of evidence comes from C/N data. Despite decreases in photosynthesis, plants that are acclimated to an excess of CO_2_ show increases in the C/N ratio [45,50]. Under N deficiency or when soil N is low, plant growth is impaired and sink activity is decreased [49,51]. Elevated CO_2_ increased leaf C/N ratio values in both Tempranillo Blanco and Tinto [13], as a consequence of N reduction, confirming previous works in different clones of Tempranillo Tinto [25,52]. Thus, data reported in the work of Kizildeniz et al. [13] for Tempranillo Blanco and Tinto confirm previous reports working with Tempranillo Tinto [25,52,53]. Irrespective of water availability and temperature, Tempranillo Blanco and Tinto acclimated to the elevated CO_2_, which indicates that the phenomenon of acclimation to elevated CO_2_ in Tempranillo grapevine is not altered by the interaction with other stress factors. This means that this phenomenon would probably occur in the future due to the increased atmospheric CO_2_ concentration, independently of whether changes in temperature and precipitation are more or less intense. Photosynthetic acclimation was more intense one week before veraison and at maturity [13]. The possible relationship between phenological period and photosynthetic acclimation is a matter that deserves further investigation in grapevine, but a lower sink demand has been hypothesized during the phenological periods in which acclimation is more severe [13].

In the literature, different causes for this photosynthetic acclimation can be found. They include (i) stomatal closure and its associated decrease in sub-stomatal CO_2_ concentration (C_i_), (ii) impaired Rubisco activity and/or (iii) reduced amount of Rubisco [27,39,41,46,47,48,49,54,55,56]. The above-mentioned causes should be, however, ruled out in Tempranillo Blanco and Tinto. First, although elevated CO_2_ closed stomata [13], confirming previous reports in several plant species [57,58], Tempranillo Blanco and Tinto sub-stomatal CO_2_ concentration (C_i_) values at maturity were much less affected by growing the plants at different CO_2_ concentration (Figure 2).

The only exception was Tempranillo Blanco grown under full irrigation at elevated temperature, in which elevated CO_2_ decreased C_i_ when compared to ambient CO_2_ (Figure 2). These rather unaltered C_i_ values in grapevine grown at elevated CO_2_ were ascribed to a large photorespiration/carboxylation ratio [24]. Second, extractable Rubisco activity remained fairly constant regardless of the treatment applied, including elevated CO_2_ (Figure 3).

Finally, in line with reports working with alfalfa [60], Rubisco concentration does not seem to be affected in Tempranillo Blanco and Tinto by the treatment of elevated CO_2_. Total soluble protein was used as proxy of Rubisco, and results showed no remarkable changes (Figure 4). Leaf water content (Appendix A) and photosynthetic pigments (chlorophyll *a + b*, Appendix A; and carotenoids, Appendix A; determined following the procedure described in Lichtenthaler [61]) did not limit photosynthesis. Elevated CO_2_, indeed, increased significantly leaf water content. Although chlorophyll and carotenoids decreased (23% on average) under elevated CO_2_, decreases of such magnitude in green, healthy leaves do not affect either leaf absorptance or photosynthesis [62,63]. Other reasons should therefore cause the photosynthetic acclimation. Photosynthetic rates are usually lowered by diffusional (stomatal and mesophyll CO_2_ conductance), photosynthetic electron transport, biochemical (different enzymes including Rubisco), and/or end-product limitations. Regarding diffusional limitations, on one hand, stomatal conductance (see above) did not limit photosynthesis. On the other hand, mesophyll conductance was reported to decrease in Tempranillo Tinto when grown under elevated CO_2_ [24]. Interestingly, and in agreement with data obtained from other species [64,65], leaf starch of CO_2_-acclimated Tempranillo Blanco and Tinto was reported to be negatively correlated to photosynthetic rates [13]. It was concluded that a product feedback inhibition cannot be excluded [13].

In the same work, the possible relationship between C/N ratios and starch in leaves was investigated [13], giving highly significant, positive relationships. Tempranillo Blanco and Tinto reached maximum values of both leaf starch concentration and leaf C/N very similarly. Furthermore, the CO_2_-mediated C/N ratio increase was very similar between varieties, 24% in Tempranillo Blanco and 29% in Tempranillo Tinto. All these findings led us to conclude that the extent of the photosynthetic acclimation does not change with the variety [13]. On the other hand, at any given leaf starch, Tempranillo Tinto had lower C/N ratios than Tempranillo Blanco. Lower leaf N concentration in Tempranillo Blanco, which increases its C/N ratio, might be related to single-nucleotide variations that occurred in the chromosome 5 of Tempranillo Blanco and that would affect amino acid metabolism [2]. It has not been explored yet if this single-nucleotide variation observed in Tempranillo Blanco causes a non-synonymous mutation or it affects a regulatory region of the gene, otherwise its functional effect would be questionable.

## 7. Conclusions

This review summarizes from the available literature differences found between Tempranillo Blanco and Tinto, in a climate change context and using fruit-bearing cuttings grown in temperature gradient greenhouses. Differences found include changes in growth, water use, grape mass, grape quality (both in technological and phenolic maturity), and in some aspects of their photosynthetic response when grown in an atmosphere of elevated CO_2_ concentration. Tempranillo Tinto consumed more water than the Tempranillo Blanco variety, and its larger consumption was associated with a larger leaf area per plant, a higher density of roots, and a higher number of stomata per unit leaf area. Elevated CO_2_ stimulated growth, but a varietal difference could be observed. When grown under elevated CO_2_, CO_2_ stimulated preferentially root (Tempranillo Tinto) and leaf (Tempranillo Blanco) growth. Differences in both technological and phenolic maturity were also observed. Within the former, sugars and pH increased. The low acidity of Tempranillo Blanco was ascribed to lower malic and tartaric acid concentrations than in Tempranillo Tinto. Within the latter, total polyphenol index was lower in Tempranillo Blanco than in Tempranillo Tinto. Causes are found in the absence of anthocyanins (resulting in green-yellow grapes) and the lower concentration of flavonols. Additionally, bunch FW and water content in berries were higher in Tempranillo Blanco when compared to Tempranillo Tinto. Photosynthetic acclimation is a phenomenon of photosynthetic down-regulation that usually occurs when plants are grown during at least some weeks at elevated CO_2_ concentration. The extent of the photosynthetic acclimation in response to elevated CO_2_ concentration was similar in Tempranillo Blanco and Tinto. Specific single-nucleotide variations affecting amino acid metabolism might be the origin of the lower levels of leaf N concentration in Tempranillo Blanco, although the functional implications of this specific mutation deserve further investigation. These findings may be used as clues for carrying out further research on the genetic differences between Tempranillo Tinto and its mutated variety Tempranillo Blanco.

Some questions, however, arise from the results summarized in this review. How different are the new features indicated in this work between Tempranillo Tinto RJ-43 clone and Tempranillo Blanco to those observed between two random Tempranillo Tinto clones? The answer is not easy. Tempranillo Tinto is an old cultivar with many available clones, many of them with strong phenotypic differences [35,67]. It is possible therefore that some of the differences found here between Tempranillo Tinto RJ-43 and Tempranillo Blanco could also be found between Tempranillo Tinto RJ-43 and any other Tempranillo Tinto clone. Then, a new question arises: How many of the described differences in Tempranillo Blanco could have their origin in the mutational event that gave place to the loss of berry color? The answer is that more research is needed. Genetic comparisons between Tempranillo Tinto and Blanco were made using clone RJ-51 as reference [2,3]. Comparing this genetic work and data reported in this review, two candidates emerge: (i) lower leaf N level and (ii) lower number of stomata per unit leaf area in Tempranillo Blanco. These candidates may be used as clues for carrying out further research on the genetic differences between Tempranillo Tinto and its mutated variety Tempranillo Blanco. In order to minimize the variability found under field conditions, the approach could be to compare different clones of Tempranillo Tinto with the only available clone of Tempranillo Blanco under fully controlled conditions, using TGG or other facilities where temperature, CO_2_, water availability, etc. can be set as desired. Once clear differences are identified, results should be confirmed in the field. With the exception of Tello et al. [3], there is no work comparing in-field results to those obtained by fruit-bearing cuttings.

## Figures and Tables

**Figure 1 plants-11-01662-f001:**
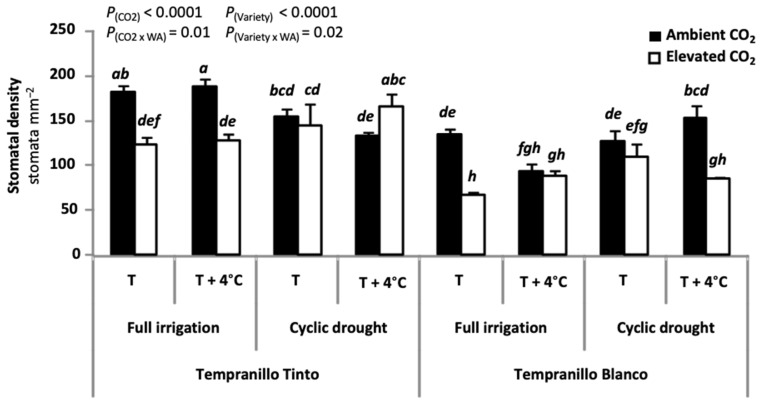
Stomatal density (stomata mm^−2^) of the leaf abaxial side (stomata were not detected in the adaxial side) at maturity in Tempranillo Blanco and Tinto grapevine. Treatments used for growing the plants were as follows: ambient CO_2_ concentration (400 ppm) versus elevated (700 ppm), ambient temperature (T) versus ambient + 4 °C (T + 4 °C), and full irrigation versus cyclic drought. Significant differences (*p* < 0.05) were based on LSD test (n = 3, mean ± SE) and are indicated by different letters. See Appendix A for optical microscope pictures. Data were obtained by T. Kizildeniz during her PhD thesis in leaf surface replicas performed using nail polish as previously described [16] and observed under an optical microscope (NIKON, YS2-H, Japan).

**Figure 2 plants-11-01662-f002:**
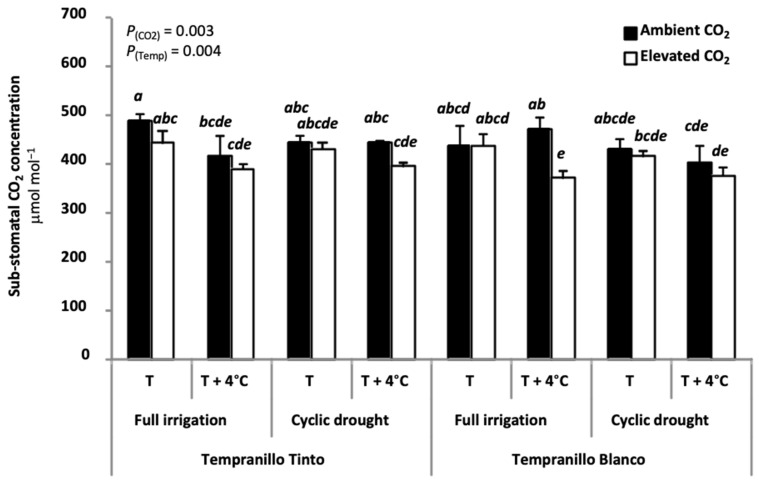
Sub-stomatal CO_2_ concentration (C_i_; µmol mol^−1^) at maturity in Tempranillo Blanco and Tinto grapevine. Treatments used for growing the plants were as follows: ambient CO_2_ concentration (400 ppm) versus elevated (700 ppm), ambient temperature (T) versus ambient + 4 °C (T + 4 °C), and full irrigation versus cyclic drought. Significant differences (*p* < 0.05) were based on LSD test (n = 5, mean ± SE) and are indicated by different letters. Gas exchange data were obtained by T. Kizildeniz during her PhD thesis and were measured at high CO_2_ concentration (700 ppm) as described in detail in Kizildeniz et al. [13].

**Figure 3 plants-11-01662-f003:**
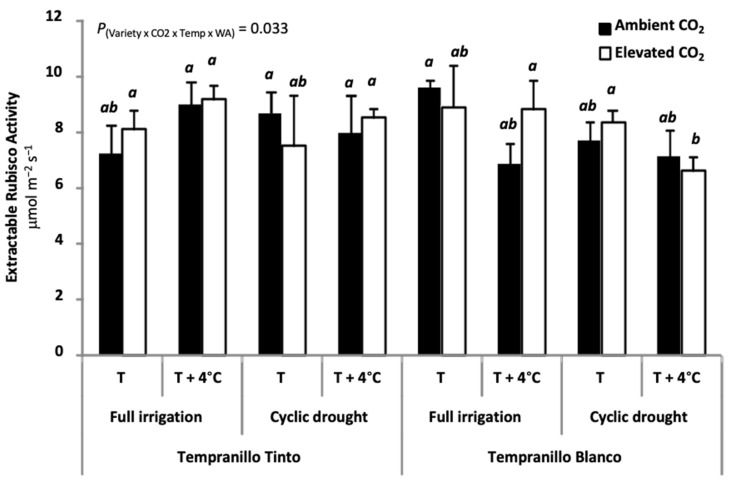
Extractable Rubisco activity (µmol m^−2^ s^−1^) at maturity in Tempranillo Blanco and Tinto grapevine. Treatments used for growing the plants were as follows: ambient CO_2_ concentration (400 ppm) versus elevated (700 ppm), ambient temperature (T) versus ambient + 4 °C (T + 4 °C), and full irrigation versus cyclic drought. Significant differences (*p* < 0.05) were based on LSD test (n = 3, mean ± SE) and are indicated by different letters. Rubisco enzyme activity data were obtained by T. Kizildeniz during her PhD thesis measuring spectrophotometrically the NADH oxidation as previously described [59].

**Figure 4 plants-11-01662-f004:**
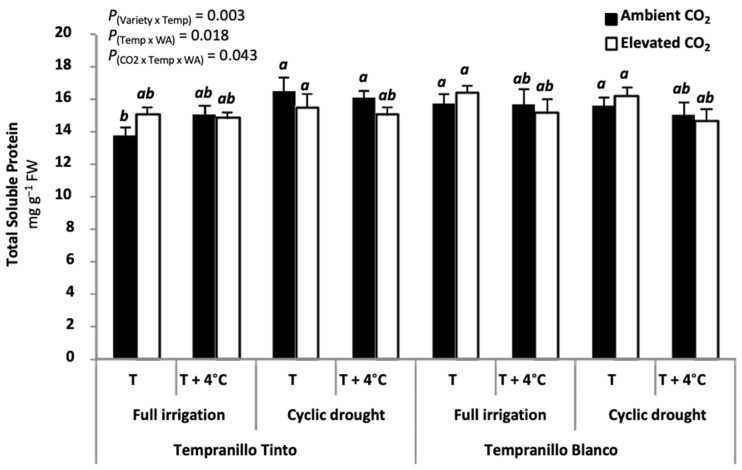
Total soluble protein (mg g^−1^ FW) at maturity in Tempranillo Blanco and Tinto grapevine. Treatments used for growing the plants were as follows: ambient CO_2_ concentration (400 ppm) versus elevated (700 ppm), ambient temperature (T) versus ambient + 4 °C (T + 4 °C), and full irrigation versus cyclic drought. Significant differences (*p* < 0.05) were based on LSD test (n = 5, mean ± SE) and are indicated by different letters. Total soluble proteins data were obtained by T. Kizildeniz during her PhD thesis using the method of Bradford [66].

**Table 1 plants-11-01662-t001:** Total flavonols (Flav) at maturity in Tempranillo Blanco and Tinto grapevine. Treatments used for growing the plants were as follows: ambient CO_2_ concentration (ACO_2_, 400 ppm) versus elevated (ECO_2_, 700 ppm), ambient temperature (T) versus ambient + 4 °C (T + 4 °C), and full irrigation (FI) versus cyclic drought (CD). Significant differences (*p* < 0.05) were based on LSD test (n = 3, mean ± SE) and are indicated by different letters. Means without letters or with common letters are not statistically different. Main factors and interactions (*p*-values) that were not significant are not shown. Data were obtained by T. Kizildeniz during her PhD thesis following the procedure described in Hilbert et al. [38].

				Flav (mg g^−1^ Skin DW)	Flav (mg g^−1^ Skin DW)		
				2013–2015	2013	2014	2015
Tinto	T	ACO_2_	FI	2.61 ± 0.94 abc	1.54 ± 0.08 bc	1.79 ± 0.28 bc	4.49 ± 0.98 ab
			CD	2.74 ± 0.76 ab	2.22 ± 0.15 a	1.76 ± 0.18 bc	4.24 ± 0.78 abc
		ECO_2_	FI	2.44 ± 0.68 abc	1.53 ± 0.32 bc	2.03 ± 0.42 bc	3.76 ± 0.61 abc
			CD	3.06 ± 0.51 a	2.30 ± 0.41 a	2.84 ± 0.65 a	4.04 ± 0.39 abc
	T + 4 °C	ACO_2_	FI	2.47 ± 0.63 abc	1.52 ± 0.05 bc	2.20 ± 0.04 ab	3.67 ± 0.47 a–d
			CD	2.08 ± 0.71 abc	1.15 ± 0.15 cd	1.61 ± 0.28 bcd	3.47 ± 0.92 a–f
		ECO_2_	FI	2.41 ± 0.56 abc	1.80 ± 0.34 ab	1.91 ± 0.20 bc	3.53 ± 0.67 a–e
			CD	2.69 ± 0.95 ab	1.94 ± 0.26 ab	1.56 ± 0.12 b–e	4.57 ± 0.91 a
Blanco	T	ACO_2_	FI	0.94 ± 0.25 c	0.53 ± 0.12 e	0.89 ± 0.18 def	1.39 ± 0.14 g
			CD	1.89 ± 0.51 abc	1.48 ± 0.36 bc	1.29 ± 0.25 c–f	2.91 ± 0.31 b–g
		ECO_2_	FI	1.14 ± 0.47 bc	0.49 ± 0.06 e	0.86 ± 0.05 def	2.06 ± 0.34 d–g
			CD	1.33 ± 0.31 abc	0.79 ± 0.12 de	1.33 ± 0.31 c–f	1.85 ± 0.30 fg
	T + 4 °C	ACO_2_	FI	1.22 ± 0.32 bc	0.64 ± 0.06 de	1.29 ± 0.12 c–f	1.74 ± 0.19 g
			CD	0.93 ± 0.24 c	0.59 ± 0.10 de	0.81 ± 0.05 ef	1.39 ± 0.24 g
		ECO_2_	FI	1.14 ± 0.45 bc	0.56 ± 0.14 de	0.83 ± 0.15 ef	2.02 ± 0.37 efg
			CD	1.32 ± 0.74 bc	0.43 ± 0.04 e	0.73 ± 0.15 f	2.79 ± 0.26 c–g
Cultivar				**0.000**	**<0.0001**	**<0.0001**	**<0.0001**
CO_2_				0.788	0.838	0.662	0.559
Temp				0.442	**0.012**	0.087	0.491
Irrigation				0.491	**0.011**	0.876	0.255
Cultivar × CO_2_				0.746	**0.017**	0.161	0.585
Cultivar × Temp				0.833	0.901	0.684	0.657
Cultivar × Irrigation				0.874	0.862	0.679	0.701
CO_2_ × Temp				0.660	0.087	**0.042**	0.088
CO_2_ × Irrigation				0.724	0.888	0.150	0.608
Temp × Irrigation				0.384	**0.001**	**0.005**	0.964
Cultivar × CO_2_ × Temp				0.916	0.545	0.295	0.939
Cultivar × CO_2_ × Irrigation				0.554	0.122	0.557	0.297
Cultivar × Temp × Irrigation				0.872	0.782	0.790	0.457
CO_2_ × Temp × Irrigation				0.562	0.252	0.781	0.123
Cultivar × CO_2_ × Temp × Irrigation				0.663	0.858	0.394	0.349

## Data Availability

Data can be obtained from authors upon reasonable request.

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
