# Peer review of "Is Tempranillo Blanco Grapevine Different from Tempranillo Tinto Only in the Color of the Grapes? An Updated Review"

_plants, 2022, doi:10.3390/plants11131662_

Round 1
Reviewer 1 Report
A review article by Kizildeniz et al. has summarized the research output and ideas on white and red Temprannillo grapevines and the differences between these varieties. The review article is well organized and well written. However, some parts of the article, primarily where the authors provide data figures, will confuse readers about whether the data has been collected specifically for this review or whether it has been adapted from the previous studies.
I suggest authors should state clearly the origin of these datasets. If these are original data sets generated for this review, please explain the details separately in the "materials and methods" section, and the review can be categorized as 'perspective articles.' If data is obtained or summarized from previous studies, mention the source in the figure legend.
The rest looks acceptable to me.
Author Response
"Please see the attachment."

Reviewer 2 Report
The manuscript ID plants_1732699 “Is white Tempranillo grapevine different from the red one only in the color of the grapes? An updated review.” by Kizildeniz and co-workers reviews the latest information regarding the differences between two grapevine cultivars: the black-berried cultivar ‘Tempranillo Tinto’ and its white-berried somatic variant ‘Tempranillo Blanco’, with an emphasis on those traits that might be useful for cultivar adaptation to novel climate conditions. In general, the manuscript is easy to read and it collects most of the available information on this topic. Nevertheless, I indicate some points that could help to improve the content of this review:
- Title
According to the VIVC database, the accepted names for the two grape cultivars analyzed in this work are ‘Tempranillo Tinto’ and ‘Tempranillo Blanco’ (https://www.vivc.de/index.php?r=passport%2Fview&id=12350 and https://www.vivc.de/index.php?r=passport%2Fview&id=25057, respectively). In fact, ‘Tempranillo Blanco’ was the name used when including this cultivar in the Spanish List of Commercial Varieties (https://www.boe.es/diario_boe/txt.php?id=BOE-A-2005-364). Therefore, authors should use these two names all along the manuscript.
Similarly, according to the OIV descriptor 225, ‘Tempranillo Tinto’ is a blue/black-berried cultivar. Consequently, authors should avoid the use of ‘red berries’ to refer to ‘Tempranillo Tinto’ berries. This comment applies to the whole manuscript.
- Abstract/Keywords
24-25. The strong differences in reproductive performance-related traits should be also cited.
3. An introduction to white Tempranillo, a “new” berry somatic variant of red Tempranillo
32. Why is the word “new” in quotes?
35. Here, the work of Martinez et al. (2006) is more appropriate than [1]:
Martínez et al. Una nueva variedad blanca para la DOCa. Rioja: el Tempranillo Blanco. XXIX Congreso Mundial de la Viña y el Vino (OIV). Logroño, junio 2006.
35. The number of ha under cultivations should be indicated.
45. These three linkage groups should be indicated.
In this section, a more extensive analysis on the previous works that compare ‘Tempranillo Tinto’ and ‘Tempranillo Blanco’ plants is needed. It includes (but not limited to) the discussion of the following works for:
- Different ampelographic features:
Balda and Martínez de Toda (2017) Variedades minoritarias de vid en La Rioja. Consejería de Agricultura. https://www.larioja.org/agricultura/es/publicaciones-agricultura/monografias/variedades-minoritarias-vid-rioja
- Different reproductive performance:
Zinelabidine et al (2021) Genetic variation and association analyses identify genes linked to fruit set-related traits in grapevine. Plant Science 306, 110875.
Tello et al (2021) Reduced gamete viability associated to somatic genome rearrangements increases fruit set sensitivity to the environment in Tempranillo Blanco grapevine cultivar. Scientia Horticulturae 290, 110497.
- Different berry color and effects on wine attributes
Martínez et al (2014) Evaluación agronómica y enológica de la variedad Tempranillo blanco (Vitis vinifera L.) y de otras variedades minoritarias blancas de la DO Ca. Rioja. I Jornada del Grupo de Viticultura y Enología, Logroño.
In addition, a Figure showing the main distinctive features of ‘Tempranillo Tinto’ and ‘Tempranillo Blanco’ will be useful in this review section.
4. The fruit-bearing cutting technique as a tool to investigate differences between white and red Tempranillo
5. The temperature gradient greenhouses simulate future climate conditions for growing white and red Tempranillo
Although the inclusion in this review of new information of the traits that differentiates ‘Tempranillo Blanco’ and ‘Tempranillo Tinto’ is of high interest, these two sections look like the conventional “Material and Methods” section of a research manuscript. I suggest authors to rewrite these two sections, reviewing (i) the different methods used to evaluate differences between ‘Tempranillo Tinto’ and ‘Tempranillo Blanco’, and (ii) the usefulness of fruit-bearing cuttings and temperature gradient greenhouses (compared to alternative methods).
L.54. Why was clone RJ-43 chosen as a control? To my understanding, if RJ-43 is not the ‘mother’ plant of the ‘Tempranillo Blanco’ line, some of the phenotypic differences observed between both genotypes could be the result of clonal differences between RJ-43 and the clone in which ‘Tempranillo Blanco’ arose, and not from the genome reshuffling indicated in this review. It should be discussed.
58-78. Is there any significant difference between the growth/success rate of ‘Tempranillo Tinto’ and ‘Tempranillo Blanco’ fruit-bearing cuttings?
100. How many cuttings per genotype and treatment combination?
126. Which organ/s were sampled?
6. Growth, water use and production of white and red Tempranillo fruit-bearing cuttings under simulated climate change conditions
140-143. Is there any difference in ‘Tempranillo Blanco’ and ‘Tempranillo Tinto’ stomata dimensions?
144-145. The list of affected loci/genes is available in Carbonell-Bejerano et al. (2017). Do this loci/genes support this hypothesis?
160. Define DW and FW. Check all the abbreviations used in the manuscript to ensure they are defined.
162. The work of Houel et al. (2013) is more appropriate than [17]:
Houel et al (2013) Genetic variability of berry size in the grapevine (Vitis vinifera L.). Australian Journal of Grape and Wine Research, 19, 208–220.
- Tempranillo white and red grape quality
Grape quality is affected by many features (see a recent review by Poni et al., 2018). The title of this section should be adapted to what is discussed:
Poni et al. (2018) Grapevine quality: A multiple choice issue. Scientia Horticulturae 234, 445-462.
178-181. Which previous experiments? Which genes?
183-184. “total polyphenol index was lower in the white variety when compared to the red one, which can be due, at least in part, to the absence of anthocyanins (resulting in white, colorless grapes) and the lower concentration of flavonols”. The lower total polyphenol index of ‘Tempranillo Blanco’ compared to that of ‘Tempranillo Tinto’ is obviously due to the absence of anthocyanins in white berries. Please, correct.
186. “Also, bunch FW and berry water content were higher in the white than in the red variety”. If FW means “Fresh weight”, this assumption contradicts previous findings (see Tello et al. (2021). Please, discuss.
8. Physiology of white and red Tempranillo: gas exchange properties and photosynthetic acclimation
213-215. Were these photosynthetic rates values significantly different between ‘Tempranillo Tinto’ and ‘Tempranillo Blanco’?
308. Does this single nucleotide variation observed in ‘Tempranillo Blanco’ cause a non-synonymous mutation? Or does it affect a regulatory region of the gene? Otherwise, its functional effect is questionable.
9. Conclusions
330-333. “Photosynthetic acclimation, a phenomenon of photosynthetic down-regulation that usually occur when plants are grown during at least some weeks at elevated CO2 concentration, was so severe in red than in white Tempranillo.” Please, rewrite.
333-334. See my previous comment (L.308).
rLast, given the experience accumulated by the authors in this topic, the following questions should be discussed in the manuscript:
- How different are the new features indicated in this work between ‘Tempranillo Tinto RJ-43’ and ‘Tempranillo Blanco’ to that observed between two random ‘Tempranillo Tinto’ clones? ‘Tempranillo Tinto’ is an old cultivar with many available clones, many of them with strong phenotypic differences (see Tortosa et al. 2016; Plant Science 251, 35–43).
- Related to my comment in L.54: How many of the described differences in ‘Tempranillo Blanco’ could have their origin in the mutational event that gave place to the loss of berry color?
- Is there any work comparing in-field results to those obtained by fruiting cuttings (in any cultivar/condition)?
TABLES
Table 1.
This table is, in fact, two independent tables. One contains data on anthocyanins, and the other on flavonols. Due to obvious reasons, anthocyanins were not evaluated in ‘Tempranillo Blanco’ berries, so only data for ‘Tempranillo Tinto’ are reported. Therefore, I suggest to eliminate this first table, or report it as an additional file.
Author Response
"Please see the attachment."

Reviewer 3 Report
This paper is an updated review of Tempranillo grapevine, namely the differences found between white and red Tempranillo, in a climate change context.
Line 160: “berry and rachis FW”, please write the FW meaning in full and “increased leaf DW” please write the DW meaning in full
Author Response
"Please see the attachment."

Reviewer 4 Report
The paper has to be rejected for the following reasons:
- It is unclear which are the new data and which are the already published data. It is not correct to entitle the paper as a review because only very recent data from the same authors are reported. The paper should only focus on new unpublished data. The paper can be resubmitted according to whatever just written.
Author Response
"Please see the attachment."

Round 2
Reviewer 2 Report
Dear authors,
Thanks for considering my suggestions for this new version. Please, consider these additional minor comments:
L. 116-118. “Dormant cuttings of Vitis vinifera L. cv. Tinto (accession T43, clone RJ-43) and Blanco…”. Indicate Tempranillo Tinto and Tempranillo Blanco.
L. 193. Please, explain what a “current temperature” is.
L. 306-312. Here, authors should state that the minor differences observed in the reproductive performace between cultivars are under the conditions assayed (fruit-bearing cuttings – greenhouse). Under field conditions, there are strong differences between the reproductive performace of Tempranillo Blanco and Tempranillo Tinto (starting from gamete viability and ending in strongly different yields).
L. 318-322. Is this hypothesis supported by the list of affected genes provided in Carbonell-Bejerano et al. (2017)?
L. 328-329. Here, authors state that Tempranillo Blanco FW from fruit-bearing cuttings is lower to that of Tempranillo Tinto, which is contrary to what observed under field conditions by Tello et al (2021), Carbonell-Bejerano et al. (2017), Balda and Martínez de Toda (2017), etc. This should be discussed in the manuscript, including the usefulness of the use of fruit-bearing cuttings to evaluate grapevine reproductive traits.
Table 1. This table does not compare Tempranilllo Tinto and Tempranillo Blanco, it should be eliminated or included as a Supplementary material.
